# Across-Shift Changes in Viable Nasal Bacteria among Waste-Incineration Plant Workers—A Pilot Study

**DOI:** 10.3390/ijerph19158984

**Published:** 2022-07-23

**Authors:** Marcin Cyprowski, Anna Ławniczek-Wałczyk, Agata Stobnicka-Kupiec, Małgorzata Gołofit-Szymczak, Rafał L. Górny

**Affiliations:** Biohazard Laboratory, Department of Chemical, Aerosol and Biological Hazards, Central Institute for Labour Protection—National Research Institute, 16 Czerniakowska Street, 00-701 Warsaw, Poland; anlaw@ciop.pl (A.Ł.-W.); agsto@ciop.pl (A.S.-K.); magol@ciop.pl (M.G.-S.); ragor@ciop.pl (R.L.G.)

**Keywords:** waste-incineration plant workers, nasal swabs, bacterial aerosols, size distribution

## Abstract

The aim of this pilot study was to assess the time-related changes in viable nasal bacteria concentrations among waste-incineration plant (WIP) workers compared to a group of office building (OB) workers outside the plant. In total, 20 volunteers participated in the study, including 14 WIP and 6 OB workers. WIP workers were divided into two sub-groups: supervisory staff (SVS) and maintenance and repair workers (MRW). Nasal swabs were collected before and after the morning work shift. Airborne bacteria were sampled with a six-stage impactor to assess the bioaerosol size distribution. The analysis showed that a significant, almost three-fold increase in nasal bacterial concentration was found only among WIP workers, and this referred mainly to anaerobic species. The load of anaerobic bacteria at the beginning of work was 12,988 CFU/mL, and after work shift 36,979 CFU/mL (*p* < 0.01). Significant increases in microbial concentrations was found only in the MRW subgroup, among non-smoking workers only. The results showed increased bacterial concentration in WIP nasal samples for as many as 12 bacterial species, including, e.g., *Streptococcus constellatus*, *Peptostreptococcus* spp., *E. coli*, and *P. mirabilis*. These preliminary data confirmed that the nasal swab method was helpful for assessment of the workers’ real-time exposure to airborne bacteria.

## 1. Introduction

The use of quantitative and qualitative analyses of the bioaerosols in different occupational settings has important practical value; however, as recent research reports show, this may be inadequate to accurately assess the impact of microbial exposure on workers’ health [1,2,3]. Scientists have been seeking biological indicators of exposure that would allow a precise description of this relationship. One of the promising methods that may help to achieve this goal is checking the upper respiratory tract for actual bacterial colonization using nasal swab [4]. The simplicity of such testing and the possibility to employ multidirectional analysis of the collected biological material make this method more and more prevalent in the assessment of workers’ exposure to bioaerosols [5].

Previous studies in this field showed that researchers have been focused mostly on the transmission of antibiotic-resistant bacterial strains in relation to the work performed. These studies referred mainly to the methicillin-resistant *Staphylococcus aureus* (MRSA) among pig farm workers [6,7], food workers (e.g., butchers, meat sellers, cooks) [8], health care workers [9], and veterinarians [10]. Another group of indicator organisms were bacterial spores of the genus *Bacillus* determined in the nasal mucosa of postal workers in the U.S. [11]. The nasal swab method has been also used in the studies on bacterial colonization of the upper respiratory tract among paper mill workers [12], health care workers [13], fuel workers [14], and pig farmers [15]. It has also been extremely successful in studying the dynamics of the time-dependent changes in bioaerosol exposure. However, these studies concerned mainly the pig farm workers [16,17,18,19]. To our knowledge, there have been no reports describing the effects of exposure to bioaerosols among workers at municipal waste-management facilities using nasal swab approach.

Currently, the most popular waste-recycling methods include composting, mechanical sorting, and recovery of waste as well as its incineration [20]. In Poland, significant changes in the methods of waste management took place over the last six years, when seven new plants for the thermal waste treatment were established. (Eight incineration plants are in operation at present.) Moreover, sorted waste has also been used as a refuse-derived fuel (RDF) for cement production. It is estimated that in Poland, about 28% of the municipal waste is subject to thermal utilization [21].

The development of new plants has resulted in the creation of new workplaces, which makes it necessary to carry out a comprehensive assessment of exposure to occupational hazards, including biological agents such as bacteria, fungi, and their toxins. The hygienic assessment of working conditions in the municipal waste incineration plants is complicated due to the complexity and variability of chemical [22] and biological pollutants emitted during the combustion process [23].

The aim of this preliminary study was to assess across-shift changes in the content of viable nasal bacteria among waste-incineration plant workers exposed to bioaerosols (study group) and a group of office workers (control group) outside the plant. The analysis included quantitative and qualitative determination of isolated strains and the size distribution of bacterial aerosols.

## 2. Materials and Methods

### 2.1. Study Design

The study was conducted in 2018 among workers at two different facilities: the waste incineration plant (WIP) and office building (OB). WIP combusted mixed municipal waste from an agglomeration with over 300,000 inhabitants. At the time of the study, the number of the plant’s employees totaled 56, including 37 maintenance and repair workers. Out of this staff, 14 workers expressed interest in participating in the project. The OB employed a total of 40 workers, and only 6 of them agreed to take part in the study.

The volunteers provided a written consent to participate, and the study protocol was approved by the Bioethics Committee of the Institute of Rural Medicine in Lublin, Poland (resolution No. 3 of 28 May 2018)

Depending on the activities performed during the work, WIP workers were divided into two sub-groups: supervisory staff (SVS), usually doing computer work, and maintenance and repair workers (MRW), including mechanics, electricians, and heavy equipment operators. The workers who were enrolled for the study worked in the morning shifts, from 7:00 a.m. to 3:00 p.m., and the measurements were taken in the middle of the working week (Wednesday). A brief description of the study population is presented in Table 1.

### 2.2. Nasal Swab Sampling

The study participants were instructed about the swabbing procedure. Nasal swabs were collected before and after the morning work shift. The sampling was performed by trained medical personnel, using sterile eSwab™ swabs (Copan Diagnostics Inc., Murrieta, CA, USA) with a nylon fiber tip, and 1 ml of liquid Amies medium, according to WHO recommendations [24]. The swabbing procedure was as follows: in order to retrieve the swab sample, the worker’s head was gently tilted back and held by the chin. The moistened end of the sterile swab was placed in the right nostril of the worker, about 2 cm deep, and the swab was vigorously wound around the walls of the nostril. The same procedure using the same swab was repeated in the left nostril. Each swab with the collected material was then placed in a test tube with Amies medium and transported to the laboratory.

### 2.3. Bioaerosol Sampling

The air-quality assessment in WIP covered three workplaces where the technological process takes place, i.e., the waste delivery hall, the vibrating screen for metal residues, and the crane control room. In OB, the sampling points were located in two office rooms.

Airborne bacteria were sampled with a 6-stage Andersen impactor (model 10-710, Graseby-Andersen, Inc., Atlanta, GA, USA), which can separate particles sized > 7/4.7/3.3/2.1/1.1/0.65 µm in diameter, in order to assess the size distribution of the bioaerosols. During the measurements, the impactor was set at a height of approx. 1.5 m above the ground. The sampling time was 5 min, the air flow rate 28.3 L/min, and the volume of the collected air sample was equal to 0.1415 m^3^. In the intervals between the sampling sessions, the impactor was disinfected and cleansed with isopropyl alcohol and then dried with a stream of hot air.

For bioaerosol sampling, the impactor was loaded with Petri plates containing the following media (all manufactured by bioMérieux, Marcy L’Etoile, France), according to Atlas [25]: tryptic soy agar (TSA) with 5% additive of sheep blood for aerobic bacteria and Schaedler agar with 5% additive of sheep blood for anaerobic bacteria. Regarding the previous authors’ experience in occupational settings concerning municipal waste [26,27], the use of mentioned growth media were appropriate in this pilot study.

### 2.4. Identification of Microorganisms

In the laboratory, the tubes with swabs were vigorously shaken for 5 min, and then, a series of 10-fold dilutions in the range of 10^−1^–10^−4^ was prepared based on a normal saline. In the next step, 200 μL of the suspension was picked and plated by surface swabbing on Petri plates with microbiological media, the same as for bioaerosol sampling.

The incubation conditions for aerobic bacteria were: 1 day (37 °C) + 3 days (22 °C) + 3 days (4 °C) and for anaerobic bacteria: 2 days (37 °C) + 2 days (30 °C). The extended incubation period for microbial samples was applied to facilitate the growth of the slow-growing strains at low temperatures [28]. Microbial colonies were counted visually. The final microbial concentration was expressed in colony-forming units (CFU) present in 1 mL of the nasal swab fluid (CFU/mL) or in 1 m^3^ of the sampled air (CFU/m^3^).

All isolated microorganisms were identified to the genus or species level. Such taxonomic analysis was based on their capability for enzymatic degradation of organic substrates through detection of the appropriate metabolites generated by the reaction. For this purpose, a set of API test kits, i.e., 20 Staph, 20 Strep, 20 NE, 20 E, 50 CHB/E, Coryne, and 20 A (bioMérieux), was applied to detect clinically important genera/species.

### 2.5. Statistical Analysis

To evaluate the findings regarding bioaerosol exposure, descriptive statistics were calculated, namely the arithmetic means (AM) with standard deviation (SD), median (Me), and the min–max range. Since the Shapiro–Wilk test revealed non-normal distribution of independent variables, the non-parametric Mann–Whitney (M-W) and Wilcoxon matched-pairs signed rank (WT) tests as well as Spearman’s rank correlation coefficient were used to confirm the statistical significance of the observed relationships. Chi-square test was used to investigate the diversity of identified microorganisms. All calculations were performed using “STATISTICA data analysis system”, version 10. (StatSoft, Inc., Tulsa, OK, USA), adopting *p* < 0.05 as statistically significant.

## 3. Results

### 3.1. Quantitative Analysis of Nasal Bacteria

The quantitative analysis of total (understand here as culturable) bacteria in nasal swabs from WIP workers (Table 2) showed an over two-fold increase in their concentrations during a work shift. This finding was statistically significant (*p* < 0.01) and related mainly to the presence of anaerobic bacteria. The median value of bacterial load before the beginning of work for all the WIP workers under study was close to 13,000 CFU/mL and after work increased to as much as 36,979 CFU/mL (*p* < 0.01). As regards the aerobic bacteria, the increase in their concentration was much smaller and statistically insignificant. However, the performed analysis showed that the type of work was the factor affecting nasal bacteria concentration among WIP workers. Significant increases in microbial concentrations could be only found in MRW subgroup. On the other hand, among SVS, these increases were either lower than among MRW, or, as in the case of aerobic bacteria, their concentrations in the nasal mucosa were even found to decrease during the work shift. The analysis also showed that the concentrations of total and anaerobic bacteria increased significantly among non-smoking workers (*p* < 0.05).

The findings obtained for the six-person group of OB workers were quite different from those noted for WIP workers. The study did not show that bacterial concentrations in the upper respiratory tract of OB workers had changed significantly as a result of work shift. However, the comparison of the median bacterial concentrations in this group revealed a trend indicating a decrease in the degree of nasal colonization by both the aerobic and anaerobic bacteria (Table 2).

### 3.2. Quantitative Analysis of Bioaerosols

The results of the quantitative analysis of bioaerosols (Table 3) showed that their levels in the plant air were an order of magnitude higher than those detected in the office building; however, these differences were not statistically significant due to the large variability of bacterial concentrations at different sites of the plant. The highest bacterial concentration in the air (7805 CFU/m^3^) was found in the waste-delivery hall and the lowest (757 CFU/m^3^) in the vibrating screen for metal residues. In the office rooms, the bacterial aerosol concentrations were more homogeneous and did not exceed the value of 163 CFU/m^3^. What distinguishes the OB environment from plant work places was the lack of anaerobic bacteria in the air of the tested office rooms.

### 3.3. Qualitative Analysis of Microorganisms

The study identified a total of 52 bacterial species belonging to 29 genera (Figure 1). The analysis of microbiota in WIP workers revealed a total of eighteen bacterial species, including eight strains of anaerobic bacteria (e.g., *Clostridium*, *Porphyromonas,* and *Streptococcus* genera) and four Gram-negative rods (e.g., *Escherichia* and *Proteus* genera). The study showed that the Gram-negative bacteria were more frequently found in MRW group (42.8%) than among SVS (21.4%); however, this difference was not statistically significant. Additionally, the presence of pathogenic *Staphylococcus aureus* was detected in 64% of samples from WIP workers. The comparison of data obtained before and after the work shift showed increased bacterial concentration in nasal samples for as many as 12 bacterial species, with the highest levels found for *S. epidermidis*, *Streptococcus constellatus*, *Peptostreptococcus* spp., *Porphyromonas asaccharolytica*, and *Clostridium beijerinckii*. Increased concentrations were also found for *E. coli* and *P. mirabilis*. The highest decreases in bacterial concentrations after work were recorded for *Actionomyces naeslundii*, *S. aureus*, and *S. hominis*.

The nasal microbiota among OB workers after the work shift comprised a total of 10 bacterial species, which was significantly less than among WIP workers (chi-square test = 4.47; *p* < 0.05). Of the identified strains, 60% belonged to *Staphylococcus* genus, including the anaerobic species of *S. saccharolyticus*, which was present in all subjects, and *S. aureus*, which was found to be carried by half of the workers. Moreover, non-sporing Gram-positive rods of *Corynebacterium* and *Brevibacterium* genera as well as rods of *Bacillus* genus were also determined. In turn, no Gram-negative rods were found in nasal swab samples. The highest concentration increase during work shift was observed for *S. epidermidis*, *S. hominis*, and *S. aureus*.

The analysis of bioaerosol samples from the waste incineration plant revealed the presence of 33 and in the office building of 10 bacterial species, and this difference was statistically significant (chi-square test = 22.7; *p* < 0.001). The taxonomic diversity of bacterial aerosols partially resembled the variety of species in the nasal mucosa of workers after work shift, but there were also numerous bacteria of *Cellulomonas*, *Bacillus*, and *Micrococcus* genera as well as mesophilic actinomycetes of *Streptomyces* genus present in high concentrations in the air only. The analysis of particle size distribution (Figure 2) showed that actinomycetes were the dominant group of bacteria, with aerodynamic diameters ranging between 0.65–1.1 µm. However, the main component of the tested bioaerosols was bacteria identified among the particles with sizes from 3.3 µm to 4.7 µm and in the case of anaerobic bacteria, when the particle diameters exceeded 7 µm.

The findings of the qualitative analysis of bioaerosol samples collected in the office building were largely consistent with the results obtained for nasal swabs. Bacteria of *Staphylococcus* genus (mainly *S. epidermidis*) prevailed in the air of the studied rooms, but bacteria of *Bacillus*, *Micrococcus*, *Aerococcus*, and *Rothia* genera were also present there. The analysis of particle size distribution (Figure 2) showed that the determined bacteria were most frequently found among the particles with aerodynamic diameters larger than 7 µm.

## 4. Discussion

The present project on the assessment of bioaerosol exposure among waste-incineration plant workers, based on the analysis of nasal swab samples collected during work shift, seems to be the first study of this kind that concerns this specific work environment. Therefore, it is difficult to find any available literature reports that could be used as reference for outcome comparison.

Our study showed for the first time that the bacterial load in the noses of waste-incineration plant workers increased considerably during work shift. Moreover, the contamination of the upper respiratory tract depended on the activities performed during the work shift. In the nasal swabs collected from workers directly involved in the thermal process of waste disposal (e.g., mechanics, electricians, heavy equipment operators), bacterial concentration increased more than in those from the supervisory staff. The workers controlling the technological process “from a distance”, were found to be less exposed to bioaerosols than the technological line workers. These findings relate to the period of one work shift in the middle of the week. Therefore, it is not known whether the changes in bacterial colonization of workers’ noses were permanent. To assess this, a measurement period of up to several days, including the weekends, would be required. The available data from pig farms suggest that these concentrations will probably decrease for a period of several hours after the end of work and then increase again the next day [17]. However, a study by Nadimpalli et al. [19] showed that a long-term work activity in a highly polluted environment contributes to a higher prevalence of the pathogenic strains in the upper respiratory tract even despite the longer breaks at work.

The analysis of the data obtained in the present study showed that bioaerosol exposure was related to the worker’s smoking habit. Although tobacco is a well-recognized source of bacteria [30], due to the risk of fire, smoking in the plant was limited to specially designated areas only. Therefore, in order to smoke a cigarette, the employees had to leave their workplace for 10–15 min. Consequently, their exposure to bioaerosols from waste processing was lower than in non-smokers. This thesis would have to be confirmed using personal sampling, as was done in the case of, e.g., pig farms [16].

As previously mentioned, similar studies have so far been carried out only among pig farmers. However, most of them have focused on analyzing the prevalence of MRSA strains. These used mostly qualitative analyses (“present vs. absent”), examining the presence of *S. aureus* bacteria in the noses of selected volunteers as a consequence of exposure in this work environment [16,19].

A different approach was presented in the report by Islam et al. [17], which also included quantitative data on the concentrations of *S. aureus* before and after work shift over one working week. This study showed that, as a consequence of occupational exposure, the median concentration of MRSA strains in the nasal mucosa at the end of work (16,000 CFU per swab) was about 60% higher than the concentrations measured in the morning (9800 CFU per swab). In contrast to those findings, Angen et al. [16], who investigated a similar work environment, noted that just after the completion of work in pig farms, the concentrations of MRSA strains in the human noses reached a level of up to 650 CFU/mL in nasal swab fluid.

In our study, the MRSA strains were not particularly notable in the waste-incineration plant. Nonetheless, as much as 60% of waste plant workers were found to be the carriers of this pathogen. A similar level (50%) in this respect was found when the office workers were examined. Available literature data indicate that this bacterial species is present in the upper respiratory tract in people of different ages and occupations. In a Dutch study [31], around 30% of children were found to be its carriers as well as 35% of medical students [32], 21% of hospital medical staff [13], and 65% of pig farmers [19]. In our study, *S. aureus* was not present in workplace air, but according to Hossain et al. [33], this pathogen is a constant component of the medical waste microbiota. Although the incineration plant under study does not intentionally receive hospital wastes, the presence of medical waste produced by the city dwellers (e.g., needles, syringes, dressings), which is incinerated as mixed waste, cannot be ruled out.

The stationary measurements carried out in two different occupational settings as well as the qualitative analysis of bacterial aerosols along with the analysis of their size distributions indicate a possible aspiration of the bacteria released during the waste-incineration process. The characteristics of the bacterial groups revealed that the changes in their load in the workers’ nasal mucosa were mainly dependent on to the prevalence of anaerobic bacteria. This discovery sheds a new light on the problem of microbiological contamination in this work environment since this group of bacteria has not yet been detected in thermal waste disposal. Given the conditions of the combustion process, most studies have focused on the bacterial contamination in relation to their growth at different temperatures rather than on their oxygen requirements [34,35,36]. The bacterial concentrations determined in WIP in the present study were similar to those described by Sabatini et al. [35]. However, recent reports indicate that the treatment of municipal waste is accompanied by a strong emission of anaerobic bacteria. Thus far, their presence has been confirmed in waste-sorting plants [1,26], in a composting plant [27], and in landfills [37]. In our study, their percentage contribution to the total bacteria pool was about 18%, which was several times less than, for example, in a waste-sorting plant [26]. Interestingly, no bacteria from this group were found in the office premises, while 10 species belonging to 8 genera were identified at work sites in the incineration plant. The presence of these species (mainly from *Clostridium* and *Peptostreptococcus* genera) indicates that the transported waste was contaminated with organic matter, most likely with food residues that have started to undergo the fermentation processes [38]. Several hours of exposure to such taxonomically complex bioaerosols during a work shift contributed to the penetration of these bacteria into the workers’ nasal cavity, which was not observed in the reference group.

Only the anaerobic species of *Staphylococcus saccharolyticus* was found in the noses of office workers, which, according to latest research, should be treated as a permanent element of the human skin microbiota [39]. It can also act as indicator organism for laryngological infections [40], in particular sinusitis [41]. However, this species to a lesser extent colonized the noses of waste-incineration plant workers, which may be associated with the higher concentrations of bacteria from other genera in this work environment. It is also likely that the activity of this species could be influenced by the presence of, e.g., *Staphylococcus epidermidis* or *Streptococcus constellatus*, which together accounted for over 30% of all bacteria in the noses of WIP workers. According to available reports, they can produce bacteriocins that inhibit the growth of other bacterial species [42]. However, this process has not been thoroughly investigated and requires further in-depth research.

Municipal waste delivered for thermal treatment was characterized by fecal contamination, as evidenced by the presence of the following genera of Gram-negative rods: *Bacteroides* [43], *Escherichia*, *Proteus* [44], and *Porphyromonas* [45]. Nasal swab analysis in our study showed that these bacteria were present only among WIP workers. A genetic analysis to determine the affinity of the isolated strains would be necessary to confirm the colonization of the workers’ upper respiratory tract by these waste-related bacteria. It obvious that molecular techniques make it possible to obtain a more accurate picture of the microbiota as compared with the cultural methods. However, due to the preliminary nature of the present study, the focus was on identifying viable bacteria that are a potential source of infection for WIP workers.

Based on the classification of harmful biological agents in workplace [29], we concluded that six species/genera belonging to risk group 2 were identified in the noses of WIP workers. These include *Clostridium* genus, whose representatives may be responsible for skin wound infections; *S. aureus*, *E. coli,* and *P. mirabilis,* which are opportunistic pathogens that can cause infections of the digestive, urinary, and respiratory systems as well as the skin; and mesophilic actinomycetes (from *Actinomyces* and *Streptomyces* genera), which show strong allergenic properties. As regards the exposure of OB workers, only two potentially harmful bacterial strains, namely *S. aureus* and *Corynebacterium* spp., were determined. Both of them may slow down wound healing. It should be also mentioned that all Gram-negative rods, which are the source of endotoxins, can pose a health threat to the incinerator workers.

The negative impact of bioaerosols on workers’ health, which is associated with the infectious or allergenic properties of bacteria, may be enhanced due to the deposition of these particles in different parts of the respiratory system. To the best of our knowledge, the size distribution analysis of the bacterial aerosols in the waste-incineration plant in our study was the second attempt of this kind after that carried out by Heo et al. [46]. In this Korean study, a gradual aggregation of bacterial cells into dust particles could be seen, reaching the highest concentration levels for particles with the aerodynamic diameters of 4.7–7.0 µm. In the present study, the phenomenon of particle aggregation was also noticed, but for both examined bacterial groups, the highest levels were observed for particles with aerodynamic diameters ranging between 3.3–4.7 µm. This finding implies that the inhaled bacteria could reach the area of the trachea and primary bronchi. Unlike in the Korean project, our study showed a 10% share of the mesophilic actinomycetes in the bioaerosol. It is very likely that a large load of the biological particles (including Gram-negative rods as well as spores of *Clostridium* and *Bacillus* genera) may have reached as far as the pulmonary bronchioles. Thus, we can assume that they may have experienced irritation of the nose and throat as well as cough and various allergic reactions.

The nasal swab analysis after work shift, especially for the anaerobic bacteria, showed that the detected bacterial groups were similar to those identified at the first impactor stage, i.e., with an aerodynamic diameter bigger than 7 µm. One of the recent clinical studies of people with COPD (chronic obstructive pulmonary disease) [47] showed that it is highly probable that the microorganisms found in nasal swabs may also reach the lungs themselves, as was shown in pulmonary sputum samples. This was observed for pathogens such as *P. aeruginosa*, *S. pneumoniae*, and *S. aureus* as well as anaerobic bacteria. Ibironke et al. [48] estimated that approximately 5% of bacterial species present in the upper respiratory tract could be also detected in sputum samples, which indicates a possible impact on lung health. Given that Gram-negative rods, which are a source of endotoxins and can reach the lower pulmonary bronchioles, are likely to induce severe inflammatory reactions in the lungs [49]. As a consequence, they may contribute to decrease of spirometric parameters, such as FEV_1_, FEF_50_, and FEF_75_, which was confirmed for incinerator workers in other studies [50,51].

## 5. Conclusions

As evidenced by the findings of this pilot study, the colonization degree of the upper respiratory tract of waste-incineration plant workers increased significantly during the work shift comparing to office workers and the changes depended on the type of occupational tasks. The nasal swab method appeared to be helpful for assessment of the workers’ real-time exposure to airborne bacteria. However, one should note that the outcomes of this preliminary research require further confirmation in a larger-scale study, with a large population of exposed workers and an extended observation time in order to investigate the persistence of detected changes. This study confirmed the previous reports that anaerobic bacteria, including those of *Clostridium* genus, play a special role in the occupational exposure of waste-management workers. The analysis of bioaerosol particle size distribution showed that they can effectively colonize all parts of the workers’ respiratory tract.

## Figures and Tables

**Figure 1 ijerph-19-08984-f001:**
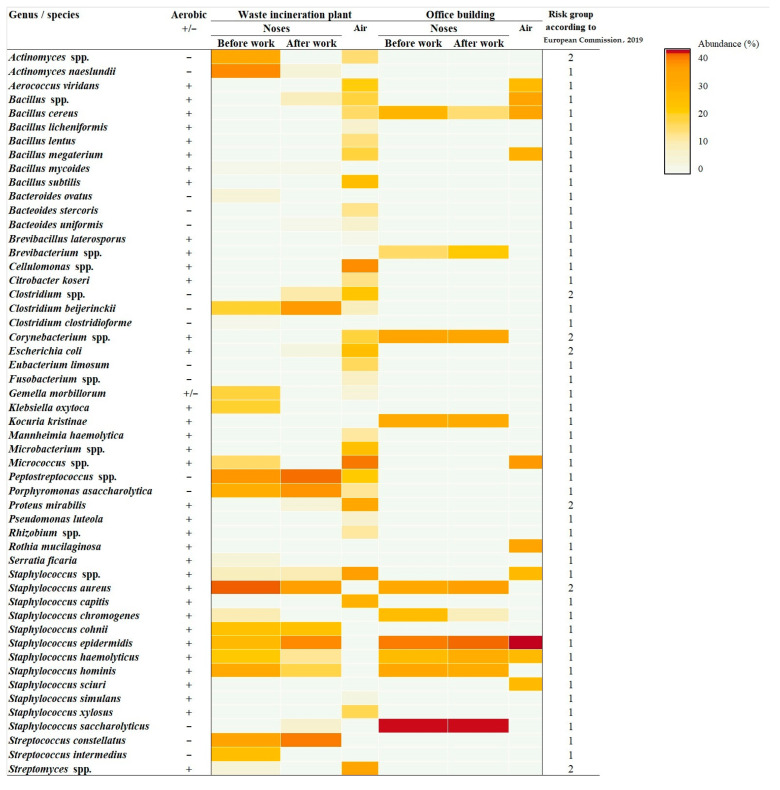
Qualitative characteristics of airborne bacteria identified in the worksites under study [29].

**Figure 2 ijerph-19-08984-f002:**
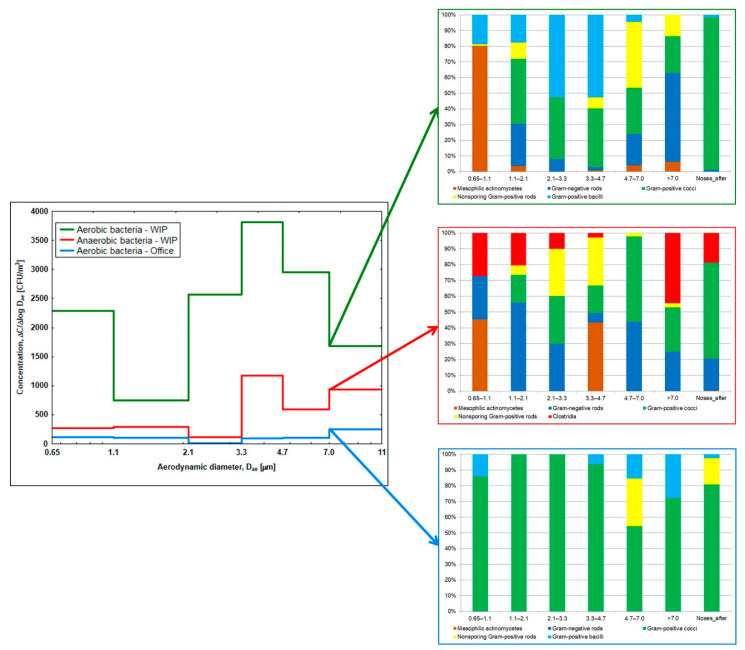
Size distribution analysis of bacterial aerosols in waste incineration plant (WIP) and office building.

**Table 1 ijerph-19-08984-t001:** Study population characteristics.

Parameter	Waste Incineration Plant (WIP)		Office Building (OB)	Total
	Supervisory Staff	Maintenance and Repair Workers	Office Workers	
Workers (*n*)	5	9	6	20
Gender: male (*n*)	4	9	6	19
Age (years) *	38.8	39.4	40.7	39.6
Current smoking (*n*)	1	2	0	3
Employment at current place (years) *	1.9	1.8	8.3	3.8

*—mean values.

**Table 2 ijerph-19-08984-t002:** Bacterial concentrations (CFU/mL) in the noses of WIP and OB workers throughout a work shift.

Study Parameter	Number of Workers (*N*)	Before Work Shift				After Work Shift				Wilcoxon Test (Z; *p*)
		AM	SD	Me	Range	AM	SD	Me	Range	
Total bacteria										
WIP workers (Total)	14	23,305	13,560	27,210	285–42,100	71,990	54,282	58,050	3900–151,000	Z = 2.67; *p* < 0.01
Supervisory staff	5	22,610	12,566	30,820	6100–32,429	75,145	65,326	53,100	3900–148,088	ns
Maintenance and repair workers	9	23,691	14,811	26,100	285–42,100	70,237	51,426	63,000	4496–151,000	Z = 2.31; *p* < 0.05
OB workers	6	25,050	16,317	19,875	10,950–55,650	25,675	19,972	16,400	10,650–61,700	ns
Aerobic bacteria										
WIP workers (Total)	14	9586	7441	10,995	0–19,652	25,969	33,680	14,500	0–120,000	ns
Supervisory staff	5	10,611	8114	12,800	1600–19,652	5532	5483	3900	0–14,000	ns
Maintenance and repair workers	9	9016	7486	9191	0–19,600	37,323	37,712	25,200	3187–120,000	Z = 2.55; *p* < 0.05
OB workers	6	12,292	8389	9400	4300–27,800	12,658	10,475	6775	5700–31,300	ns
Anaerobic bacteria										
WIP workers (Total)	14	13,719	8903	12,988	216–32,909	46,020	44,880	36,979	0–148,088	Z = 2.67; *p* < 0.01
Supervisory staff	5	11,998	5428	12,777	4500–18,900	69,612	64,615	50,905	0–148,088	ns
Maintenance and repair workers	9	14,675	10,544	13,200	216–32,909	32,913	25,386	31,000	1309–68,771	Z = 2.07; *p* < 0.05
OB workers	6	12,758	7985	10,475	6550–27,850	13,017	9637	9625	4950–30,400	ns

*N*, number of samples; AM, arithmetic mean; SD, standard deviation; Me, median.

**Table 3 ijerph-19-08984-t003:** Bacterial aerosol concentrations (CFU/m^3^) in worksites under study.

Study Parameter	*N*	AM	SD	Me	Range	U Test (M-W)
Total bacteria						
WIP	3	3387	3849	1598	757–7805	ns
OB	2	138	35	138	113–163	
Aerobic bacteria						
WIP	3	2755	2902	1464	722–6079	ns
OB	2	138	35	138	113–163	
Anaerobic bacteria						
WIP	3	632	949	134	35–1726	ns
OB	2	0	0	0	0	

*N*, number of samples; AM, arithmetic mean; SD, standard deviation; Me, median; U test (M-W), the Mann–Whitney U test.

## Data Availability

The data presented in this study are available on request from the corresponding author.

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
