# Peer review of "Across-Shift Changes in Viable Nasal Bacteria among Waste-Incineration Plant Workers—A Pilot Study"

_ijerph, 2022, doi:10.3390/ijerph19158984_

Round 1
Reviewer 1 Report
- Write medium composition you used,
- Indicate whether you used a colony counter to count the number of microbial colonies or was this done visually?
- Was the use of one growth medium appropriate for the appearance of all microbes in the sample, or the better for you used more than one medium to be suitable for the growth of the largest number of microbes?
Author Response
Reviewer #1: (marked in the text in blue)
- Write medium composition you used – The Authors referred to the "Handbook of Microbiological Media" (Atlas RM, 1997), which describes the exact composition of the microbiological media used. The new reference was given the number 25.
- Indicate whether you used a colony counter…or was it done visually – The Authors added a sentence in the methodology description "Microbial colonies were counted visually".
- Was the use of one growth medium appropriate… – at the end of subsection 2.3, the Authors added a sentence presenting their position on this issue.
Reviewer 2 Report
I have completed the review on ijerph-1789682. The authors conducted a meaningful study on workers in waste incineration plants. The topic of this study is in the Aim & Scope of this journal. The authors used reliable methods and received relevant approval. Therefore, I think this manuscript is suitable for publication in IJERPH after revisions.
1. Line 100-103, since the absence of figures, the description is confusing.
2. For some experimental designs, without references, a more detailed interpretation is needed to ensure reliability.
3. The authors wrote the good Results section, but the Discussion section needs to be improved. First, I suggest a section discussion, which is not difficult for the authors. Second, any completed discussion needs to be summarized and reflected in the conclusions.
4. Abstract and conclusions, please cite more quantitative data, rather than vague views.
Author Response
Reviewer #2: (marked in the text in yellow)
- Line 100-103… the description is confusing – The Authors agree that sampling point numbers may be misleading as they have not been used anywhere else. They have been removed to improve the quality of the text.
- “For some experimental designs, without references, a more detailed interpretation is needed to ensure reliability” – In the Authors' opinion, the presented comment of the Reviewer is imprecise in order to make changes in the text.
- The authors wrote the good Result, but the Discussion section needs to be improved… – The Authors agree that some of the paragraphs in the Discussion section may have a different order, which increases the quality of the chapter. Several of them have been re-positioned.
- Abstract and conclusions… – The Authors have modified some of the sentences contained in the Conclusions section, as well as slightly in the Abstract.
Reviewer 3 Report
I congratulate the authors for the work and manuscript. Manuscript ID ijerph-1789682 "Across-shift changes in viable nasal bacteria among waste incineration plant workers" by Cyprowski and collaborators is well written and the work is clearly presented. The statistical methods for data analysis are appropriate and figures and tables present all the relevant data. The work has limited scope and relatively small number of studied participants but is an important contribution for the understanding of the bacterial colonization degree of the upper respiratory tract of waste incineration plant workers, which was shown to impact the concentration of airborne bacteria according to the type of occupational tasks. The authors recognize the limitations of the study and outline the need for larger scale studies to confirm the observations that can guide occupational health and safety training to minimize the adverse effects of bioaerosol exposure in waste incineration plants. I recommend the publication of the manuscript in its current form, after a small correction in line 287: replacing "depended" with "dependent".
Author Response
Reviewer #3: (marked in the text in purple)
Line 287… replacing “depended” with “dependent” – The change was applied in the text.
Round 2
Reviewer 2 Report
My concerns are addressed.